# Pan-cancer analysis reveals cooperativity of both strands of microRNA that regulate tumorigenesis and patient survival

Ramkrishna Mitra [1], Clare M. Adams [1], Wei Jiang[2], Evan Greenawalt[1] & Christine M. Eischen [1]✉

Recently, both 5p and 3p miRNA strands are being recognized as functional instead of only one, leaving many miRNA strands uninvestigated. To determine whether both miRNA strands, which have different mRNA-targeting sequences, cooperate to regulate pathways/ functions across cancer types, we evaluate genomic, epigenetic, and molecular profiles of >5200 patient samples from 14 different cancers, and RNA interference and CRISPR screens in 290 cancer cell lines. We identify concordantly dysregulated miRNA 5p/3p pairs that coordinately modulate oncogenic pathways and/or cell survival/growth across cancers. Down-regulation of both strands of miR-30a and miR-145 recurrently increased cell cycle pathway genes and significantly reduced patient survival in multiple cancers. Forced expression of all four strands show cooperativity, reducing cell cycle pathways and inhibiting lung cancer cell proliferation and migration. Therefore, we identify miRNA whose 5p/3p strands function together to regulate core tumorigenic processes/pathways and reveal a previously unknown pan-cancer miRNA signature with patient prognostic power.

[1] Department of Cancer Biology, Sidney Kimmel Cancer Center, Thomas Jefferson University, Philadelphia, PA, USA. [2] Department of Biomedical Engineering, College of Automation Engineering, Nanjing University of Aeronautics and Astronautics, Nanjing 211106, China. ✉email: christine.eischen@jefferson.edu

MicroRNA (miRNA) are small non-coding RNA that regulate gene expression by binding target messenger RNA (mRNA) and typically inhibiting translation[1,2]. Although individual miRNA are reported to have many critical roles in tumorigenesis[1,2], much remains unknown. Moreover, there are >8300 miRNA identified in humans[3], and each precursor miRNA consists of two mature miRNA strands (5p and 3p) that contain different mRNA-targeting sequences[4,5]. Recent evidence has shown the historical belief that one strand is degraded during miRNA biogenesis may not be correct[4,5]. Instead, depending on the tissue/cell type, developmental stage, or disease state both miRNA strands can be present or have altered expression[4,6,7]. In cancer in particular, genomic instability, epigenetic modifications, and alterations in transcription factors impact miRNA loci[1,8,9], which can change the expression of precursor miRNA, resulting in altered levels of both the 5p and 3p strands[7]. However, due to the study of typically only one miRNA strand, the function of half of all miRNA has not been explored, missing the likely significant contributions many have to human cancers.

A limited number of studies have investigated the functions of both miRNA strands of specific miRNA in specific cancers[10–20]. For example, miR-31-5p and miR-31-3p are both upregulated in oral squamous cell carcinoma, but one counteracts the oncogenic function of the other[17]. The 5p and 3p strands of miR-582 and miR-28 are downregulated in bladder and colorectal cancer, respectively[15,16]. However, forced expression of miR-582-5p and miR-582-3p inhibited bladder cancer cell proliferation and tumor growth[15], whereas forced expression of miR-28-5p, but not miR-28-3p, reduced colorectal cancer cell proliferation[16]. In miR-17 transgenic mice, both the 5p and 3p strands are overexpressed and the mice are predisposed to hepatocellular carcinoma[18]. Moreover, forced co-overexpression of miR-17-5p and 3p strands increased prostate cancer xenograft growth and invasion[19]. Therefore, miRNA have complex mechanisms of actions and the interplay between the 5p and 3p strands of individual miRNA in cancers remains largely unexplored.

As many cancer types rely on fundamental oncogenic pathways, a core set of mature miRNA may exist that are consistently dysregulated across cancer types and drive pan-cancer tumorigenesis by recurrently targeting and dysregulating genes in these critical pathways[21]. Currently, it remains unclear whether both strands of endogenous miRNA function cooperatively to regulate core oncogenic processes and pathways during tumorigenesis across cancers. To investigate this, we leveraged genome-scale RNA interference (RNAi) and CRISPR-based loss-of-function screens from hundreds of cell lines, and thousands of multi-omics profiles from 14 different cancer types available in The Cancer Genome Atlas (TCGA), and verified results with independent data sets and biological experiments. We systematically identified miRNA 5p/3p pairs that coordinately dysregulate oncogenic pathways across cancer types and determined that both miRNA strands of two top predicted miRNA cooperate to control cancer cell proliferation and migration, and constitute a core miRNA signature that correlates to patient survival across cancer types.

## Results

**miRNA 5p/3p pairs cooperatively modulate cancer cell growth.** With the analysis of 396 human and 47 mouse miRNA expression profiles across different cell types, generated by Functional Annotation of the Mammalian Genome project[22], we observed global concordance in 5p/3p expression changes. The observed concordance was significantly ($P < 10^{-74}$, Wilcoxon rank-sum test) higher compared with the background miRNA pairs, which were selected randomly from different precursor miRNA

(Fig. 1a). We obtained consistent results by analyzing 5238 TCGA miRNA expression profiles across 14 different cancer types (Fig. 1b; Supplementary Fig. 1 and Supplementary Table 1). For each cancer type, we analyzed 45−847 tumor and normal samples that had both miRNA and mRNA expression profiles. We identified 78-196 (average = 135) significantly dysregulated (1.5-fold-change with Benjamini–Hochberg (BH)[23] adjusted $P < 0.05$) mature miRNA from each cancer type (Supplementary Fig. 2). Among those identified, 9–42 miRNA pairs (average = 23) originated from the same precursor miRNA. Irrespective of whether the 5p/3p pairs had significant dysregulation in cancer, their expression fold-changes had high concordance (Spearman's $\rho = 0.74$; correlation $P = 5.89 \times 10^{-289}$) (Fig. 1c and Supplementary Fig. 3).

We investigated whether these concordantly dysregulated miRNA 5p/3p pairs work together to regulate specific biological functions. Genome-scale si/shRNA or CRISPR-based loss-of-function screens for 290 diverse cancer cell lines were available for the 9 cancer types we studied here (Methods section, Supplementary Table 2)[24–26]. This massive-scale resource provided an unprecedented opportunity to understand the relationship between miRNA expression and cell viability/growth from miRNA regulating genes that reduce the viability/growth of specific cancer cells upon their inactivation. A gene with a lower score in the screening data indicates the given cell line is more dependent on that gene for its survival/growth. We predicted cancer-specific miRNA-target gene sets and determined, using the gene set enrichment analysis (GSEA)[27] embedded into R package WebGestaltR[28], which target gene sets were significantly (FDR < 0.1) overrepresented at the bottom of the ranked gene list in the cancer cell lines of that cancer type (Methods section; Fig. 1d). The results indicate the regulating miRNA of those gene sets may negatively impact the growth and/or survival of the corresponding cancer cell lines.

We identified 71 miRNA from RNAi and 80 miRNA from CRISPR screening data with 66 being the same miRNA from both screens whose expression potentially inhibits growth and/or survival of cancer cell lines. We examined dysregulation patterns of these miRNA in the tumor tissues in TCGA of the same cancer types as the cell lines. We determined that 98% and 100% of the miRNA derived from the RNAi and CRISPR screens, respectively, had significant downregulation in the specific tumor type compared with the corresponding normal tissue (Fig. 1e). We also determined expression changes of the core enrichment genes in the selected gene sets with significant normalized enrichment scores (NES (FDR < 0.1)). From TCGA data, we observed that 99% and 100% of the genes identified from RNAi and CRISPR screens, respectively, had significant upregulation in the corresponding tumor types of the cell lines (Fig. 1e). Collectively, these results suggest that our identified cell survival/growth-inhibiting miRNA are downregulated during tumorigenesis, leading to elevated expression of their survival/growth-inducing target genes, and thereby, increasing cancer cell survival and/or growth (Fig. 1e).

Combining the results from both screens, we identified 70 miRNA that had significant dysregulation (1.5-fold-change, BH adjusted $P < 0.05$) in ≥5 TCGA cancer types. As our results suggest overexpression of these miRNA may induce cell death or inhibit cell growth, we focused on the most relevant miRNA by determining which had consistent downregulation across cancers (Fig. 1f). We identified 46 miRNA downregulated at least twice as frequently as they were upregulated, suggesting their downregulation may be selected for in the cancer cells. The same five miRNA (miR-28, miR-30a, miR-139, miR-143, miR-145) in the RNAi and CRISPR screening data showed both strands were downregulated across cancer types, indicating their

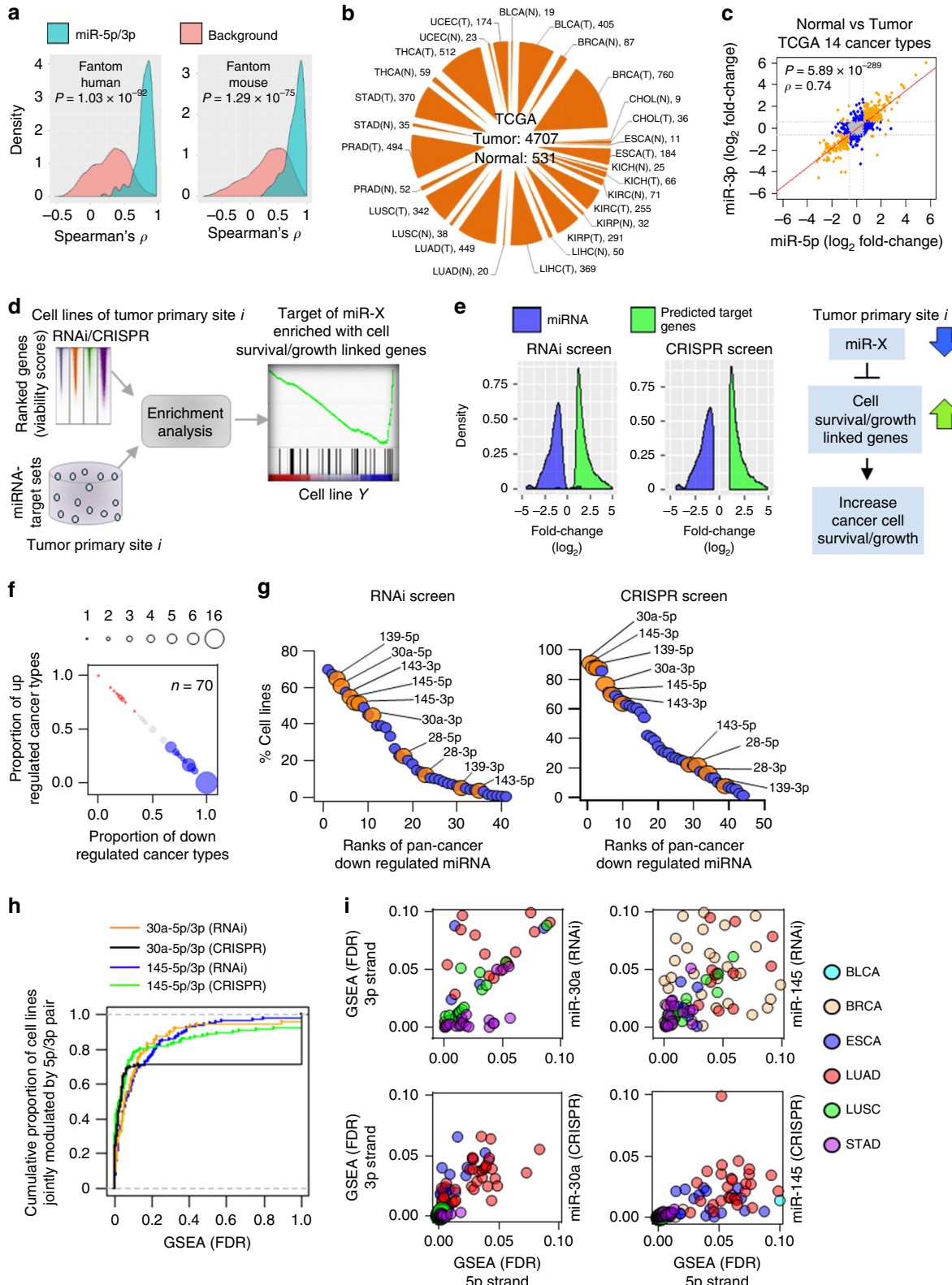

downregulation may have allowed for the survival/growth of the cells that originated from these cancer tissues (Fig. 1g, Supplementary Tables 3 and 4). The data showed one strand of miR-30a and miR-145 only had <1.5-fold higher cell survival/growth association compared with the other strand, suggesting both strands may function together. Our data also suggest both strands of miR-30a and miR-145 potentially co-modulate

survival/growth of a higher proportion of cancer cell lines compared with the other three miRNA 5p/3p pairs.

We determined from the RNAi screening data that 45–61% of the cell lines were negatively impacted by either the 5p or 3p strand of miR-30a or miR-145 (Fig. 1g). For the CRISPR screening data, the range was 70–91% (Fig. 1g). These cell lines correspond to eight cancer types (BLCA, BRCA, ESCA, LIHC,

**Fig. 1 Associations between miRNA 5p/3p concordant expression changes and cancer cell viability/growth. a** Density plots showing distribution of 5p/3p expression correlations (green) compared to the background miRNA pairs (red). *P*-values determined by two-tailed Wilcoxon rank-sum test. **b** Number of TCGA normal (N) and tumor (T) samples by cancer type used for analysis. **c** Scatterplot with regression line, Spearman correlation test score ($\rho$) and *P*-value (two-tailed) indicates degree of global concordance observed in expression fold-changes of 5p/3p pairs across 14 cancer types compared with the corresponding normal samples. Each dot represents fold-change of one 5p/3p pair in one cancer type. Significant differential expressions of both, one, and no strands denoted by orange, blue, and gray dots, respectively. **d** Integrative approach to predict whether individual miRNA target set is enriched with cell survival/growth-associated genes. **e** Density plots showing distribution of survival/growth-associated miRNA and gene expression changes in the tumor tissues compared with the respective normal tissues. Schematic diagram indicates potential association between miRNA downregulation and induced cell survival/growth. **f** Pan-cancer ($\geq 5$ cancer types) dysregulated miRNA ($n = 70$) were plotted. Blue and red dots indicate a trend of consistent down and upregulation, respectively, across the cancer types. Gray dots indicate random up- or downregulation. The proportion of down- or upregulations was indicated on the *x*- and *y*-axis, respectively. Size of individual dots correlates with number of miRNA. **g** List of miRNA (blue and orange circles) that were predicted to be associated with cancer cell viability/growth from RNAi or CRISPR screening data. *y*-axis indicates percentage of the cell lines with which these miRNA had association. Orange dots represent the 5p and 3p strands of a miRNA that both modulate cancer cell survival/growth. **h** Empirical cumulative distribution of cell lines (*y*-axis) whose survival/growth might be coordinately regulated by the indicated miRNA 5p/3p pairs, as determined by the GSEA with 10% FDR (*x*-axis). **i** Distribution of cancer cells of indicated cancer type whose survival/growth may be regulated by both miR-30a-5p and 3p (left) or both miR-145-5p and 3p (right) as determined from RNAi (top) and CRISPR (bottom) screening data. Dots represent cancer cell lines.

LUAD, LUSC, STAD, UCEC) in TCGA in which miR-30a and/or miR-145 strands were downregulated (Fig. 1g and Supplementary Tables 3 and 4). Of the cell lines whose survival/growth were potentially modulated by miR-30a or miR-145, we investigated which lines showed association from both strands. We determined the 5p and 3p strands of miR-30a coordinately associated with the survival/growth of 65% and 69% of the cell lines in the RNAi and CRISPR screening data, respectively, that correspond to 4 cancer types (ESCA, LUAD, LUSC, STAD) (Fig. 1h, i). For miR-145, both strands coordinately associated with the survival/growth of 63% and 77% of the cell lines in the RNAi and CRISPR screening data, respectively, that correspond to 6 cancer types (BLCA, BRCA, ESCA, LUAD, LUSC, STAD) (Fig. 1h, i). Combining the results from both screens, we determined that both strands of miR-30a and miR-145 may coordinately regulate the survival/growth of 79% and 63% of the cell lines, respectively. Collectively, we identified a set of miRNA whose both 5p and 3p strands are associated with cancer cell survival/growth, and the associations are reproducible in two different platforms. Specifically, this analysis revealed that downregulation of miR-30a and miR-145 5p/3p pairs may be necessary for pan-cancer cell survival and/or growth.

**miRNA 5p/3p pairs recurrently alter pathways across cancers.** Given the lack of understanding of whether and the extent to which both strands of individual miRNA can cooperate to regulate specific cellular pathways that contribute to biological phenotypic changes in cancer cells, we developed a computational framework to identify pathways that were significantly impacted due to miRNA 5p/3p concordant dysregulation. We identified pan-cancer dysregulation of 26 miRNA 5p/3p pairs that were significantly dysregulated in $\geq 5$ cancer types compared to corresponding normal samples (Supplementary Table 5). Ten had a trend of global downregulation and the remaining had global upregulation (Fig. 2a). This analysis revealed there are specific miRNA 5p/3p pairs that are coordinately up- or downregulated across different malignancies, suggesting their combined functions may be critical and thus, selected for during tumorigenesis.

We performed pathway enrichment analysis on potential targets of the above pan-cancer dysregulated miRNA by determining which have altered expression in cancer compared with the corresponding normal samples and an inverse expression association with the potential regulating miRNA (Fig. 2b). We hypothesize this strategy may capture miRNA-mediated systems-level biological changes selected for during tumorigenesis. Genes potentially regulated by the 5p or 3p strand or both in a specific cancer type were used to determine in which pathways in the

Kyoto Encyclopedia of Genes and Genome (KEGG)[29] database they were significantly (BH adjusted hypergeometric test $P < 0.05$) enriched. For 14 different cancers, we identified 178 KEGG pathways (5p/3p pair–pathway–cancer combinations = 1252) that were altered in one or more cancer types potentially due to dysregulation of at least one 5p/3p pair (Fig. 2b). We scored individual 5p/3p–pathway pairs to statistically determine the pathways that were coordinately regulated by both strands with high confidence (Methods; Fig. 2c, d).

We determined the proportion of the target genes in each pathway that were potentially co-targeted by both 5p and 3p strands of the specific miRNA. From the pool of pathways that were coordinately regulated by individual 5p/3p pairs, the observed distributions prominently clustered the miRNA into three groups. Four miRNA 5p/3p pairs, including miR-145, co-target a low proportion (<11%; average fraction of predicted co-targets present in the specific miRNA-regulated pathway pool) of their target genes (Fig. 2e, extremely left-skewed density curves), indicating they preferentially target different genes in the same pathway. Three miRNA 5p/3p pairs, including miR-17, co-target a high proportion (54–59%) of their target genes (Fig. 2e, extremely right-skewed density curves), even though they do not have identical seed sequences. The remaining 14 miRNA 5p/3p pairs, including miR-30a, co-target 19–43% of the genes in the same pathways (Fig. 2e). Therefore, 5p and 3p strands of the identified miRNA can co-target the same genes or different genes in the same pathways, and are less likely to compete with each other compared to miRNA in the same family that have identical seed sequences.

We investigated the pathways that had recurrent up- or downregulation potentially due to down- or upregulation of the specific 5p/3p pairs, respectively. Among the 498 high-confidence 5p/3p–pathway pairs (5p/3p pair–pathway–cancer combinations = 761; Supplementary Table 6), we identified 20 that recurrently appeared in at least 50% of the cancer types where corresponding regulating 5p/3p pairs had concordant dysregulation (Fig. 2f, g and Supplementary Table 6). The top five recurrently associated 5p/3p–pathway pairs were comprised of two upregulated miRNA that recurrently downregulated three pathways (miR-17-5p/3p–Proteoglycans in cancer, miR-17-5p/3p–Rap1 signaling, and miR-93-5p/3p–Vascular smooth muscle contraction) and two downregulated 5p/3p pairs (miR-30a-5p/3p and miR-145-5p/3p) that recurrently upregulated cell cycle pathway genes in at least 2/3rd of the cancer types in which the 5p/3p pairs were dysregulated (Fig. 2g).

It was previously reported that the miR-17-5p/3p pair cooperatively regulates *TIMP3*, a member of the proteoglycans

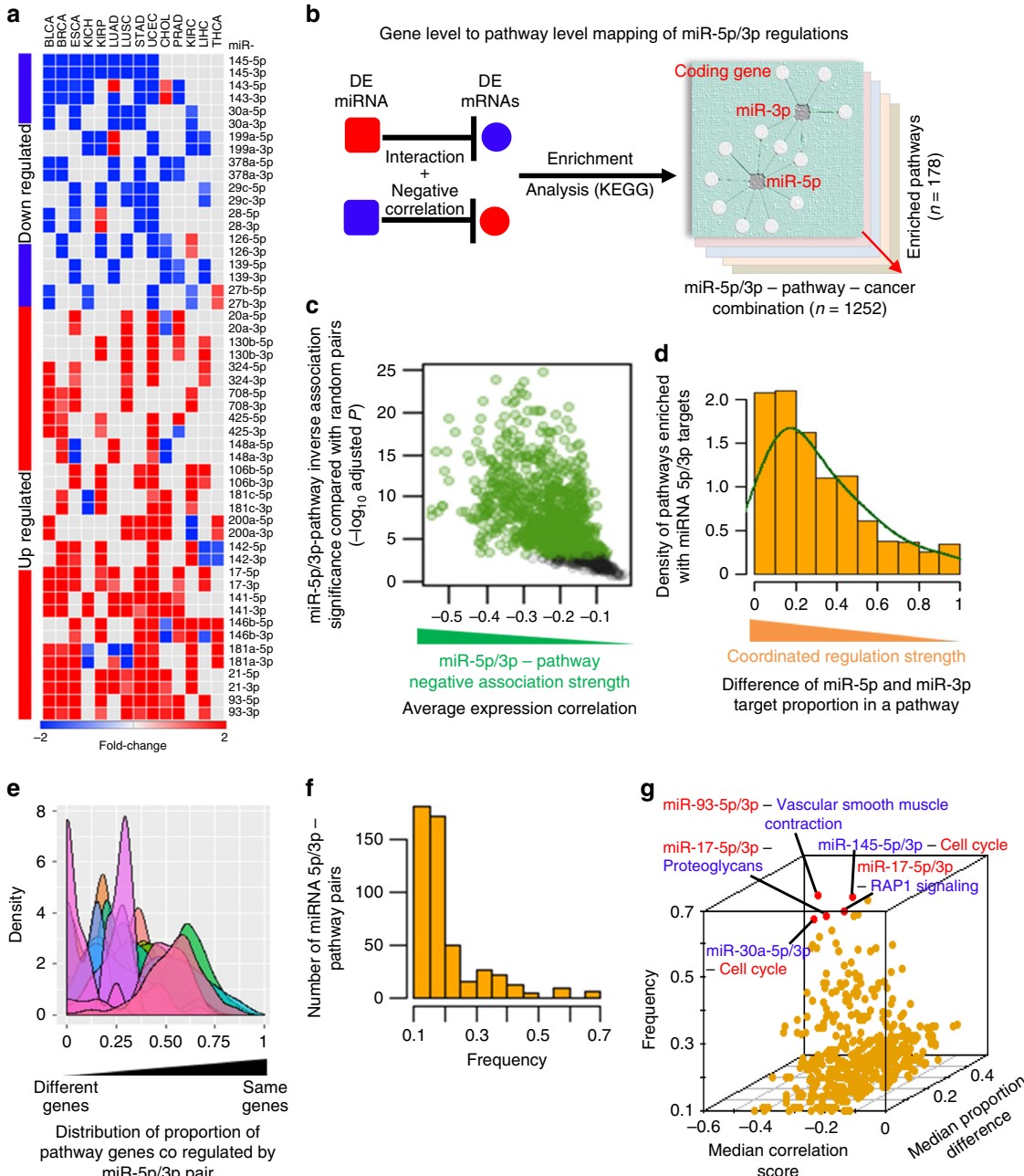

**Fig. 2 Identifying pan-cancer dysregulated miRNA 5p/3p pairs and their recurrent associations with oncogenic pathways across cancer types. a** Heat map represents pan-cancer dysregulated 5p/3p pairs. Cancer types with red or blue color represent significant up- or downregulation, respectively, of the indicated mature miRNA. **b** Overview of approach used to identify miRNA 5p/3p regulated KEGG pathways in cancer. **c, d** Determining high-confidence 5p/3p pair–pathway associations based on whether **c** they had strong (green dots) inverse associations (x-axis) or not (black dots) comparing with the random pairs (y-axis); each dot represents one miRNA-pathway pair, and **d** the pathway genes were coordinately regulated by the 5p/3p pair. **e** Determining which 5p/3p pairs coordinately regulate pathways with a trend of co-targeting the same genes (x-axis, toward 1) or different genes (x-axis, toward 0) in the same pathway. Each density curve represents one 5p/3p pair and its potentially regulated pathways across cancer types. **f** Determination of pathways that were recurrently regulated by miRNA 5p/3p pairs across the cancer types. x-axis denotes frequency (number of cancer types with 5p/ 3p–pathway association divided by the number of cancer types the regulating 5p/3p pair had concordant dysregulation). **g** Three-dimensional scatter plot summarizes inversely associated (x-axis), coordinately regulated (y-axis) 5p/3p–pathway pairs that recurrently (z-axis) associated with cancer types where the corresponding regulating 5p/3p pairs had concordant dysregulation. Top five associations are indicated (red dots); red and blue words denotes up- and downregulation in cancer, respectively, compared to normal samples.

in cancer pathway. miR-17-5p/3p-mediated suppression of *TIMP3* may contribute to prostate cancer progression, invasion, and metastasis[19,30,31]. We determined that elevated levels of the miR-17-5p/3p pair recurrently suppressed the reported target gene and pathway across multiple cancers (Supplementary Fig. 4

and Supplementary Table 5). Our results indicate the miR-17-5p/ 3p pair may coordinately induce pan-cancer cell migration and metastasis by suppressing the proteoglycans pathway. For miR-93, both strands were reported elevated in invasive pituitary adenoma compared with pituitary adenoma tissue, indicating

both strands may be dysregulated concordantly in tumorigenesis[32]. For miR-145 and miR-30a 5p/3p pairs, analysis of RNAi and CRISPR screening data and miRNA-pathway interactions data across multiple cancer types showed they may regulate the survival/growth of numerous cancer cell lines by concordantly modulating the expression of cell cycle pathway genes.

**Elevated miR-145 or 30a strands suppress cell cycle pathways**. To further analyze whether our predicted miRNA-mediated pathway regulations captured the contribution of both 5p and 3p strands of individual miRNA, we evaluated the two top predicted miRNA-pathway associations (miR-145-cell cycle and miR-30a-cell cycle). With the analysis of microarray expression profiles (GSE48414)[33] from the Gene Expression Omnibus database[34], we determined significant downregulation of both miR-145-5p (3.12-fold-change; $P = 2.52 \times 10^{-16}$; t-test) and miR-145-3p (4.20-fold-change; $P = 2.64 \times 10^{-21}$; t-test) in the lung adenocarcinoma patient cohort compared to normal lung tissue (Fig. 3a). Furthermore, analyzing another independent data set (GSE107008), we determined transcriptome-wide mRNA expression changes induced by overexpression of either the miR-145-5p or 3p strand in LUAD and ESCA cancer cell lines. Using GSEA with the gene sets available in the KEGG or Reactome[35] databases, we determined that overexpression of miR-145-5p or miR-145-3p resulted in suppression of pathways involved in cell cycle progression (Fig. 3b). Cell cycle, DNA replication, and mitotic cell cycle were among the top ten downregulated pathways that significantly decreased (FDR = 0) with increased expression of miR-145-5p or miR-145-3p in both LUAD and ESCA cancer cell lines.

From the independent microarray expression profiles, we also determined significant downregulation of both miR-30a-5p (4.23-fold-change; $P = 1.0 \times 10^{-19}$; t-test) and miR-30a-3p (5.97-fold-change; $P = 7.08 \times 10^{-25}$; t-test) in lung adenocarcinoma patient samples compared to normal lung tissue (Fig. 3c)[33]. Furthermore, using quantitative reverse transcription PCR (qRT-PCR) we confirmed significant downregulation of both miR-30a-5p ($P = 1.98 \times 10^{-6}$; t-test) and miR-30a-3p ($P = 2.60 \times 10^{-8}$, t-test) in our own lung adenocarcinoma patient cohort compared to normal lung tissue (Fig. 3d). Similarly, we determined levels of miR-30a-5p and miR-30a-3p were significantly ($P < 6.63 \times 10^{-4}$, t-test) reduced across a panel of lung adenocarcinoma cell lines compared to a normal bronchial epithelial cell line (Fig. 3e).

To gain insight into regulatory mechanisms of individual miRNA strands, we performed whole transcriptome RNA-sequencing expression profiling following transfection of miR-30a-5p or miR-30a-3p mimics into A549 lung adenocarcinoma cells and evaluated transcriptome-wide mRNA expression changes compared to samples with a negative control RNA. Using GSEA, based on KEGG and Reactome databases, we determined several cell cycle-linked pathways, including cell cycle, mitotic cell cycle, DNA replication, mitotic prometaphase, and mitotic M-M/G1 phase were in the top ten pathway list that were significantly (FDR < 0.05 for KEGG and FDR = 0 for Reactome) downregulated in both miR-30a-5p and miR-30a-3p transfected lung adenocarcinoma cells (Fig. 3f).

The cell cycle pathway was predicted to be regulated by miR-145-5p/3p in 6 different TCGA cancers (BLCA, ESCA, LUAD, LUSC, STAD, UCEC) and by miR-30a-5p/3p in four cancers (ESCA, LUAD, LUSC, STAD). We assessed variation of cell cycle pathway activation levels in individual tumor samples of these cancer types using single-sample gene set enrichment analysis (SSGSEA)[36]. We determined that individual patients with higher cell cycle activation signals had significantly lower expression of miR-145-5p, miR-145-3p, miR-30a-5p, and miR-30a-3p compared to patients with a lower cell cycle activation signal of the

above cancer types, except UCEC (Supplementary Fig. 5a, b). Therefore, using independent lung cancer validation data, we verified decreased levels of individual strands of both miR-145 and miR-30a in multiple cancers. Furthermore, using perturbation data, we verified that elevated expression of these individual mature strands results in suppression of cell cycle pathways.

**miR-30a strands hinder lung cancer proliferation and movement**. As previous reports confirmed increased miR-145-5p and miR-145-3p expression decreases proliferation and movement of lung adenocarcinoma[37], lung squamous cell carcinoma[38], and bladder cancer[39] cell lines, we focused on miR-30a[28]. Our RNA-seq data revealed significantly downregulated genes (≥2-fold-change with BH adjusted $P < 0.05$) either in miR-30a-5p or miR-30a-3p transfected lung adenocarcinoma cells compared to the cells transfected with negative control RNA. These genes were significantly enriched, as determined by the WebGestalt tool[28], with critical gene ontology biological processes that induce cell growth and movement (Fig. 4a, Supplementary Table 7). Upregulated genes were enriched with the gene ontology terms that suppress cell movement (Supplementary Table 8). These results suggest elevated miR-30a-5p and miR-30a-3p may suppress LUAD cell proliferation and movement. To evaluate the biological effects of miR-30a-5p and 3p on lung cancer cell growth, we separately overexpressed these miRNA in two lung cancer cell lines (A549 and H1993) that had reduced endogenous levels of these miRNA strands (Fig. 3e). Increased levels of either miR-30a-5p or 3p at two different concentrations resulted in decreased lung cancer cell growth compared to the control RNA in both lung cancer lines (Fig. 4b and Supplementary Fig. 6). To test the biological effects of miR-30a on cell movement, we overexpressed each strand individually or control RNA and performed transwell migration assays. As previously reported, miR-30a-5p inhibited cell migration in A549 cells[40]. Overexpression of miR-30a-3p also inhibited cell migration, but to a greater extent than miR-30a-5p, indicating both strands independently control cell movement (Fig. 4c). These data further demonstrate that both strands of miR-30a contribute to cancer cell proliferation and migration.

**Both miR-145 and 30a 5p/3p pairs cooperate in lung cancer**. Since pathways can be modulated by changes in gene expression due to genetic alterations (e.g., copy number) or epigenetic modification (e.g., methylation), we investigated whether miR-30a and miR-145-mediated cell cycle pathway modulation in 14 TCGA cancer types were influenced by such modifications. We conducted a multivariate regression analysis to determine miRNA-mediated mRNA expression changes accounting for the noise from copy number changes and DNA methylation (Fig. 5a). For LUAD, we identified predicted target mRNA that had significant negative associations (BH adjusted regression $P < 0.05$) with miR-30a-5p, 30a-3p, 145-5p, and 145-3p. We determined that each miRNA strand significantly suppressed a subset of the previously reported gold-standard common essential genes[41]. These predicted target essential genes were significantly more depleted in cells with elevated regulating miRNA compared with the remaining genes ($P < 3.3 \times 10^{-9}$; two-tailed Kolmogorov–Smironov test; Fig. 5b and Supplementary Fig. 7). Using the Rank product statistic, we identified miRNA-target association recurrence across the 14 cancer types with statistical significance (BH adjusted Rank product $P < 0.05$; Fig. 5a)[42,43]. These recurrently and inversely associated genes were significantly enriched with cell cycle pathways for both miR-30a and miR-145 (Fig. 5c). The results suggest pan-cancer suppression of cell cycle pathway genes by miR-145 and miR-30a 5p/3p pairs are independent from DNA copy number and methylation changes.

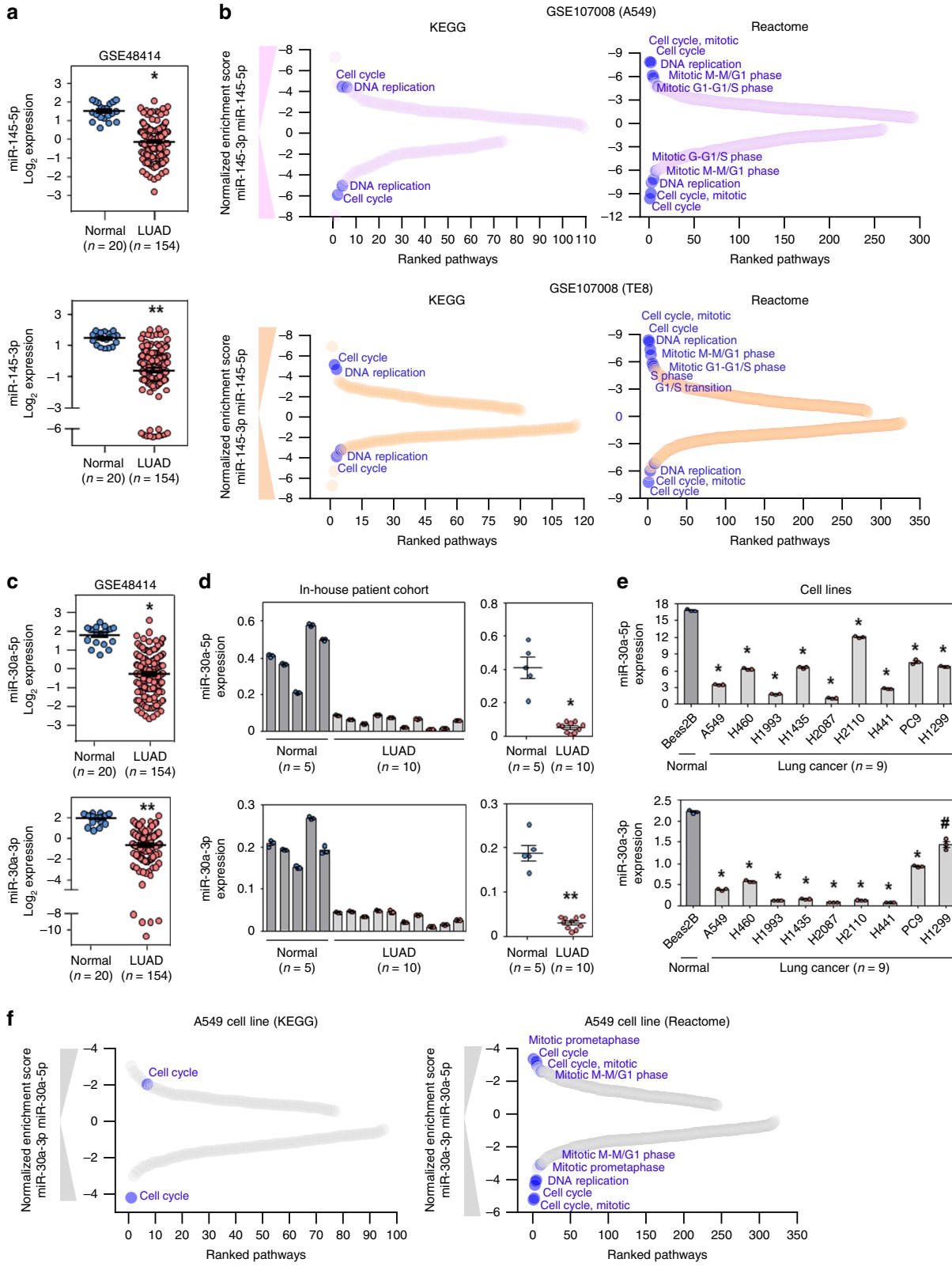

To estimate the impact of 5p/3p combined-mediated regulation of gene expression, we stratified patient samples from individual cancer types into lower and higher groups based on median expression of miR-30a-5p and miR-30a-3p. If both strands had either lower expression or higher expression in a set of samples, we denoted those samples as low or high, respectively. The remaining samples were denoted as mixed type. Notably, we observed cell cycle regulating genes recurrently associated with both miR-30a-5p and 3p had higher expression in the low-sample group and lower expression in the high sample group across cancer types (Fig. 5d, e). There was intermediate expression in the mixed group (Fig. 5e). Therefore, alteration of both strands of miR-30a may have a greater impact on cell cycle pathway genes than a single strand.

**Fig. 3 Individual strands of miR-145 and miR-30a independently modulate cell cycle-associated pathways. a, c** Microarray to measure expression differences of miR-145-5p (*$P = 2.52 \times 10^{-16}$; **a** top), miR-145-3p (**$P = 2.64 \times 10^{-21}$; **a** bottom), miR-30a-5p (*$P = 1.0 \times 10^{-19}$; **c** top), and miR-30a-3p (**$P = 7.08 \times 10^{-25}$; **c** bottom) in an independent cohort of lung adenocarcinoma patient samples ($n = 154$) compared to normal ($n = 20$) lung tissue; two-tailed $t$-tests. **b, f** GSEA demonstrated downregulated pathways in miR-145-5p or miR-145-3p transfected A549 cells (**b** top) or TE8 cells (**b** bottom), and miR-30a-5p or miR-30a-3p transfected A549 cells (**f**), compared to the control cell line. Pathways were ranked ($x$-axis) on the basis of negative normalized enrichment scores ($y$-axis). Cell cycle-associated pathways that are within the top ten pathway list, are indicated and were significantly enriched (FDR < 0.05). **d** qRT-PCR (in triplicate) for miR-30a-5p (top) and miR-30a-3p (bottom) expression in patient samples of lung adenocarcinoma and normal lung tissue. Individual samples (left) and mean expression (right) shown; *$P = 1.98 \times 10^{-6}$, **$P = 2.60 \times 10^{-8}$, two-tailed $t$-tests. **e** qRT-PCR (in triplicate) for miR-30a-5p (top) and miR-30a-3p (bottom) levels in lung adenocarcinoma lines and a normal bronchial epithelial line; #$P = 6.62 \times 10^{-4}$, *$P < 6.05 \times 10^{-6}$, two-tailed $t$-tests. For qRT-PCR (**d** and **e**), miRNA expression was normalized to endogenous small RNA *RNU6B* and presented as $2^{-\Delta Ct}$. Data are presented as mean values ± SEM in **a** and **c–e**. Source data are provided as a Source Data file.

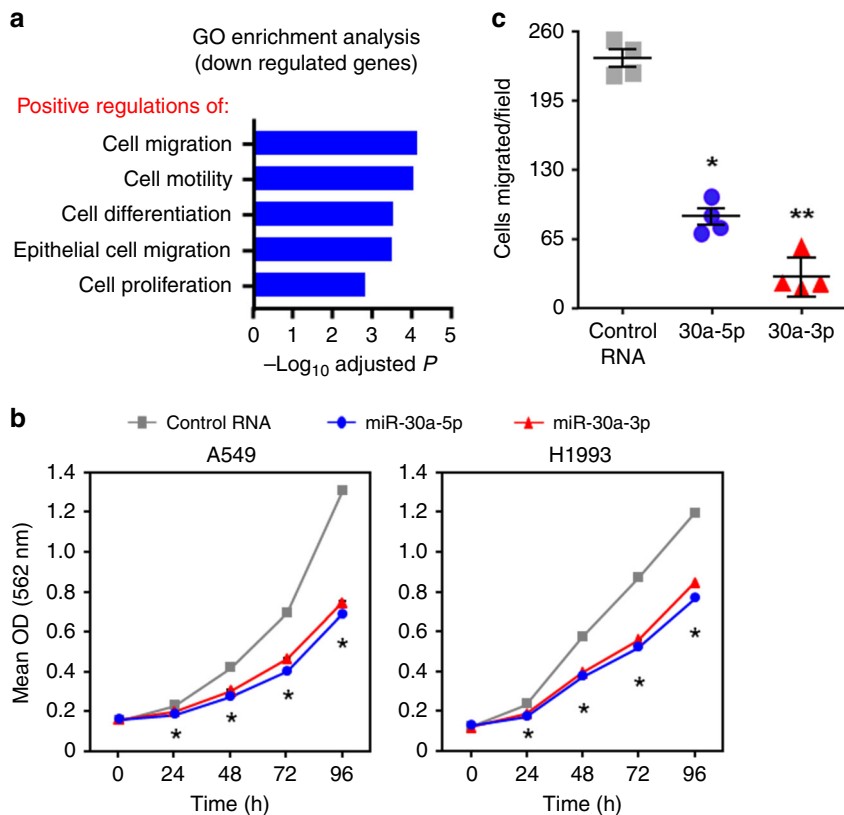

**Fig. 4 Increased levels of miR-30a-5p and miR-30a-3p decreased lung cancer cell growth and migration. a** Gene Ontology enrichment analysis based on genes that were significantly downregulated by either miR-30a-5p or miR-30a-3p determined from our RNA-seq data (GSE142695). Biological processes that were significantly enriched with downregulated genes are indicated. **b** Lung adenocarcinoma lines were transfected with 100 nM miR-30a-5p mimic, miR-30a-3p mimic, or a negative control RNA. MTT assays were performed at the indicated intervals in quadruplicate. Each assay was performed at least four independent times for both cell lines; a representative experiment is shown. For A549, *$P < 2.43 \times 10^{-3}$; for H1993, *$P < 2.91 \times 10^{-6}$; two-tailed $t$-tests (comparing 5p or 3p mimic to control RNA); error bars are within the symbols. **c** A549 cells transfected with miR-30a-5p mimic, miR-30a-3p mimic, or control RNA and transwell migration assays performed. Results from one representative experiment from two independent experiments performed is shown; *$P = 1.11 \times 10^{-5}$, **$P = 3.15 \times 10^{-6}$, two-tailed $t$-tests (comparing 5p or 3p mimic to control RNA, respectively). Data are presented as mean values ± SEM in **b** and **c**. Source data are provided as a Source Data file.

To evaluate the biological consequences when both strands of miR-30a are dysregulated, we simultaneously overexpressed miR-30a-5p and miR-30a-3p in A549 and H1993 lung cancer cell lines. There was a further decrease in cell growth when both 5p and 3p were increased together compared to when each was overexpressed alone, indicating a cooperative effect (Fig. 6a and Supplementary Fig. 8a). Moreover, the combination of both miR-30a-5p and miR-30a-3p at a lower concentration resulted in either a similar effect (H1993 cells) or a greater negative effect (A549 cells) on cell growth compared to when either miR-30a-5p or miR-30a-3p were

overexpressed individually at twice the concentration (Fig. 6b). To test the effects on cell migration, A549 cells were transfected with one or both strands of miR-30a. Transwell migration assays showed that both miR-30a-5p and miR-30a-3p together had a cooperative negative effect on lung cancer cell migration (Fig. 6c). These data show that concordant downregulation of both miR-30a strands work together to increase cancer cell growth and migration, which would be predicted to negatively impact patients.

To investigate potential cooperativity between 5p and 3p strands of both miR-30a and miR-145, we simultaneously

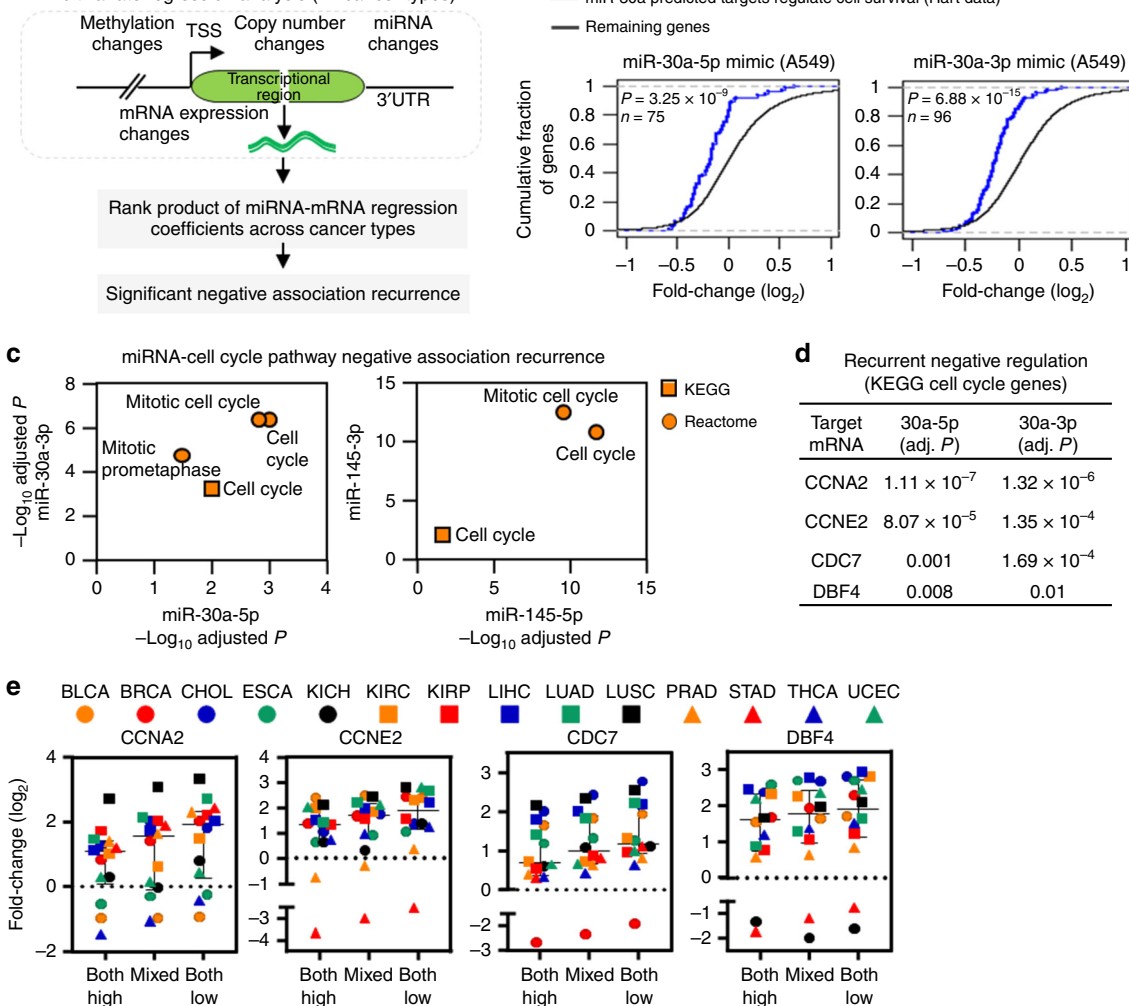

**Fig. 5 Cooperative effects of miRNA 5p/3p-mediated gene regulation in cancer cell proliferation and movement. a** Identifying miR-30a and miR-145 5p/3p-mediated gene expression changes across 14 cancer types. A linear regression function was modeled to predict miRNA–mRNA association after controlling for the impact of DNA methylation and gene copy number changes. The rank product statistic was employed to aggregate the miRNA–mRNA regression coefficients across the 14 cancer types and predict significant miRNA–mRNA-negative association recurrence (BH adjusted $P < 0.05$). **b** Expression changes of miRNA-target essential genes after transfection of the miR-30-5p or 3p mimics in A549 lung adenocarcinoma cells (the miRNA perturbation profile from our RNA-seq data, GSE142695). The miRNA-essential gene pairs were obtained from regression analysis in TCGA LUAD. Indicated $P$-values were determined from two-tailed Kolmogorov–Smirnov test. $n$ denotes number of predicted target essential genes of the corresponding miRNA. We extracted a common essential gene list from the Hart laboratory (http://hart-lab.org/). **c** Pathway enrichment analysis to determine pathways that are enriched with recurrently associated genes of the corresponding regulating miRNA indicated on the x- and y-axis. **d** Recurrent association strength of four cell cycle pathway-associated genes with predicted binding sites for both miR-30a-5p and miR-30a-3p. Adj. $P$ denotes BH adjusted $P$-value derived from rank product test statistic. **e** Fold-change (log2) of the expression of the four genes in the indicated sample categories across the 14 cancer types. High or low: Samples with higher or lower expression of both miR-30a-5p and 3p. Mixed: Remaining samples. Data are presented as median with interquartile range. Source data are provided as a Source Data file.

overexpressed all four strands in A549 and H1993 lung cancer cell lines. Using two different concentrations, we observed that cell growth was significantly reduced in both cell lines when all four miRNA strands were overexpressed compared to overexpression of both the 5p and 3p strands of miR-30a or both the 5p and 3p strands of miR-145 (Fig. 6d and Supplementary Fig. 8b, c). Notably, combining all four strands at a lower concentration (25 nM each) had a greater negative impact on cell growth than both strands of either miRNA combined at a higher concentration (50 nM each) (Fig. 6d). Similarly, in colony formation assays, there were fewer colonies of A549 lung cancer cells overexpressing all four miRNA strands compared to A549 cells overexpressing both 5p and 3p strands of either miR-30a or miR-145 (Fig. 6e). Cooperativity was also observed among all four

miRNA strands in migration assays, as migration of A549 cells was further reduced when both 5p and 3p strands of both miRNA were overexpressed compared to overexpression of the 5p and 3p strands of miR-30a or miR-145 (Fig. 6f). Collectively, these data show cooperation between both strands of miR-30a and miR-145 impact tumor cell proliferation and migration.

**Novel miRNA signature links to pan-cancer patient survival.** Given that miR-145 and miR-30a 5p/3p pairs had pan-cancer downregulation in association with overlapping oncogenic biological processes, they may constitute an oncogenic miRNA signature[44,45]. To evaluate whether expression of these miRNA correlate with patient overall survival, we averaged the expression

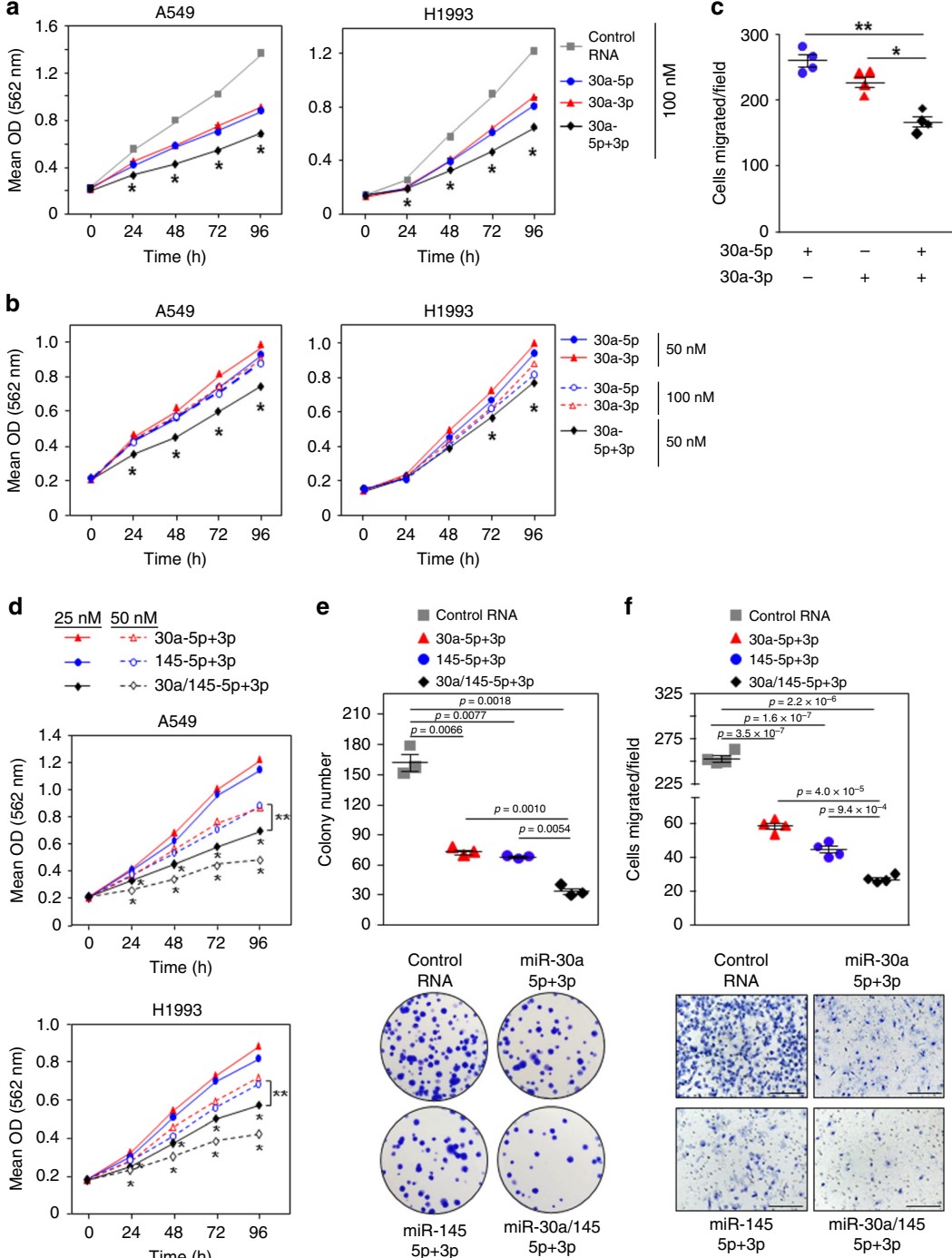

**Fig. 6 Cooperation of all four strands of miR-30a and miR-145 in cancer cell growth and movement.** Lung adenocarcinoma lines transfected with miR-30a-5p, miR-30a-3p, miR-145-5p, and/or miR-145-3p miRNA mimic, or control RNA at the indicated concentrations. Control RNA was also added to equalize the total amount of RNA transfected. **a**, **b**, **d** MTT assays performed at intervals, in quadruplicate. Each assay was performed 2–4 independent times for both cell lines and one representative experiment is shown. For **a**, A549 *$P < 6.18 \times 10^{-5}$, H1993 *$P < 2.36 \times 10^{-2}$ (comparing 5p + 3p to 5p or 3p alone); for **b**, A549 *$P < 9.21 \times 10^{-6}$, H1993 *$P < 1.91 \times 10^{-2}$ (comparing 5p + 3p to 5p or 3p alone at both concentrations); for **d**, A549 *$P < 1.69 \times 10^{-4}$ and **$P < 7.02 \times 10^{-5}$, H1993 *$P < 4.27 \times 10^{-4}$ and **$P < 2.82 \times 10^{-3}$ (*, comparing 30a-5p/3p+145-5p/3p to 30a-5p/3p or 145-5p/3p at both concentrations and **, comparing 25 nM 30a-5p/3p+145-5p/3p to 50 nM 30a-5p/3p or 145-5p/3p); two-tailed t-tests. **c**, **f** Transwell migration assays were performed 2 (**c**) or 4 (**f**) independent times and results from one representative experiment are shown for each. For **c**, *$P = 2.3 \times 10^{-3}$ and **$P = 2.0 \times 10^{-4}$; for **f**, P-values indicated; two-tailed t-tests. Representative images shown in **f** (x10 magnification; scale bars, 200 μm). **e** Colony formation assays were performed two independent times, in triplicate, and a representative experiment and images (no magnification) are shown. P-values indicated; two-tailed t-tests. Data are presented as mean values ± SEM; error bars are within the symbols in **a**, **b**, and **d**. Source data are provided as a Source Data file.

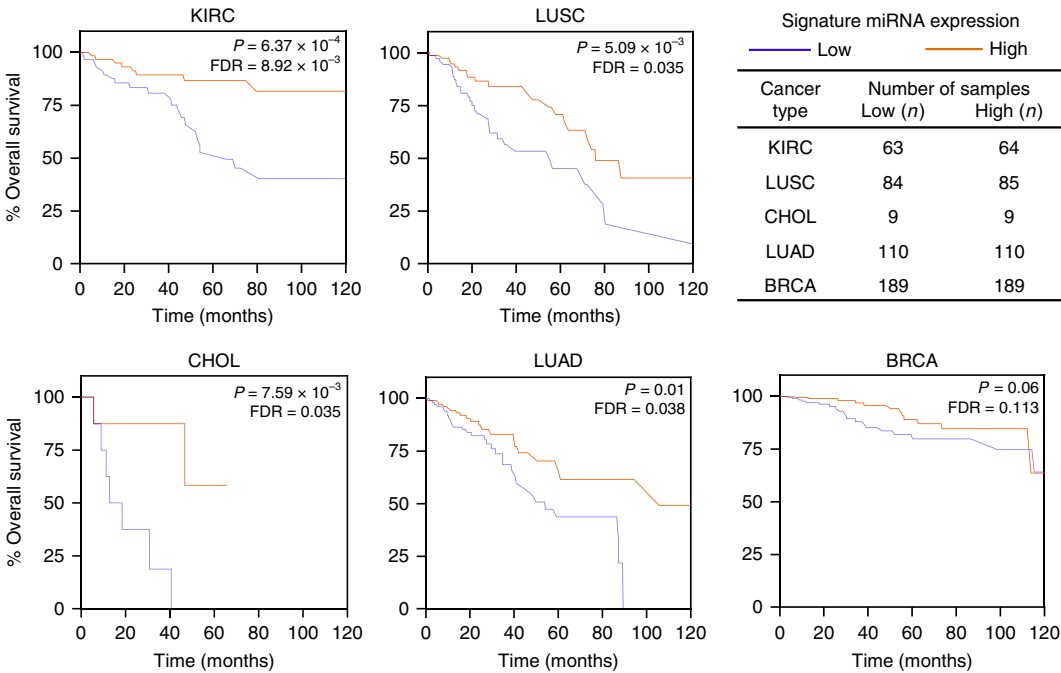

**Fig. 7 A four-miRNA expression signature correlates with patient overall survival across diverse cancer types.** Kaplan–Meier plots of overall survival according to a survival predictor based on the average expression of miR-145-5p, miR-145-3p, miR-30a-5p, and miR-30a-3p. For each cancer type, patient samples were ranked according to the average expression of the four miRNA and divided into four quartiles. Lowest (blue lines) and highest (orange lines) quartile groups were used for Kaplan–Meier survival analyses; statistical significance was measured using log-rank test. False-discovery rate (FDR) values were determined.

of miR-145-5p, miR-145-3p, miR-30a-5p, and miR-30a-3p for each TCGA patient. For a specific cancer type, we ranked the patients according to the average miRNA expression and divided them into four equal quartiles. Univariate survival analysis showed that patients with low signature miRNA expression (lowest quartile group) had significantly reduced overall survival compared with patients that had high signature miRNA expression (highest quartile group) for five cancer types: KIRC ($P = 6.37 \times 10^{-4}$; log-rank test, hazard ratio [HR] = 0.27), CHOL ($P = 7.59 \times 10^{-3}$, HR = 0.09), LUSC ($P = 5.09 \times 10^{-3}$, HR = 0.45), LUAD ($P = 0.01$, HR = 0.48), and BRCA ($P = 0.06$, HR = 0.51) (Fig. 7 and Table 1). FDR adjusted survival $P$-values shown in Fig. 7. For LUAD and LUSC, two aggressive cancer types, the median survival decreased by 51.5 months (54.1 vs. 105.6) and 20.9 months (55.2 vs. 76.1), respectively, when the miRNA levels were low. We conducted the same analysis considering the expression of a single strand or both strands of miR-145 or miR-30a. We observed a significantly ($P = 2.51 \times 10^{-3}$; Wilcoxon rank-sum test) stronger association between reduced overall survival and downregulation of all four mature strands together compared with the association observed between poor overall survival and downregulation of a single strand or both strands of miR-145 or miR-30a (Supplementary Table 9). Subsequent multivariate cox regression analysis considering miRNA signature expression, patient age, tumor grade, and tumor stage information (if available) determined the expression of these miRNA independently correlated with poor overall survival with statistical significance ($P < 0.05$, Wald test; HR < 0.56; Table 1). Evaluation of the other miRNA we identified, which cooperatively regulate the same critical oncogenic pathways (Supplementary Table 6), showed that none had an equivalent or superior survival association as that of the miR-145/30a 5p/3p combination, indicating this miRNA signature is unique. These results indicate the expression of the four strands of miR-145 and miR-30a constitute a signature that is a predictive indicator of overall survival in

patients across diverse cancer types. Altogether, the data reveal that miR-145-5p, miR-145-3p, miR-30a-5p, and miR-30a-3p are coordinately downregulated by multiple cancers due to their regulation of core processes that inhibit tumorigenesis and tumor progression that significantly contribute to patient survival.

**Discussion**
Despite miRNA sequencing data showing miRNA biogenesis produces significant quantities of mature miRNA from both the 5p and 3p strands from miRNA duplexes[4,5], very little is known about their combined functions and regulatory mechanisms in cancer and other diseases. Here, we identified 26 miRNA 5p/3p pairs that had concordant dysregulation across multiple cancer types, indicating they may be selected for during tumorigenesis. We have carried out a comprehensive, rigorous analysis of human transcriptomic, genomic, epigenetic, and gene perturbation data across 14 TCGA cancer types to gain insight into the roles of these miRNA 5p/3p pairs in cancer. We established links between frequently dysregulated miRNA 5p/3p pairs and tumorigenesis in two different ways, adding significant power to our analysis. Firstly, we exploited cell viability/growth screens that established essential gene signatures, and linked those genes with potentially regulating miRNA by developing a novel computational framework. Utilizing RNAi and CRISPR, two independent screening platforms, we determined reproducible associations between miRNA 5p/3p dysregulation and modulation of cancer cell viability/growth across cancers. Secondly, we conducted a rigorous data analysis that incorporated pathway information in the dysregulated miRNA and mRNA networks across cancer types to determine specific miRNA 5p/3p pairs that may coordinately regulate the same pathways that contribute to tumorigenesis. We derived previously unidentified, high-confidence miRNA 5p/3p-mediated pathway regulations in 14 cancer types and made them publicly available, allowing for further exploration of the candidate miRNA 5p/3p–pathway interactions in TCGA data. We also

**Table 1 Analysis of 10-year overall survival in patients of diverse TCGA cancer types.**

| Cancer type | Variables | Univariate | | | Multivariate | | |
|---|---|---|---|---|---|---|---|
| | | HR[a] | 95% CI[b] | P-value[c] | HR | 95% CI | P-value[d] |
| KIRC | Age | 1.1 | 0.54–2.25 | 0.80 | | | |
| | Tumor grade[e] | 4.17 | 1.45–11.96 | 3.89E-03 | 2.35 | 0.80–6.92 | 0.12 |
| | Tumor stage[f] | 10.19 | 3.90–26.63 | 4.61E-09 | 8.38 | 3.16–22.22 | 1.93E-05 |
| | miRNA expression[g] | 0.27 | 0.12–0.60 | 6.37E-04 | 0.29 | 0.12–0.66 | 3.35E-03 |
| LUSC | Age | 0.74 | 0.42–1.30 | 0.30 | | | |
| | Tumor stage | 1.56 | 0.82–2.97 | 0.17 | | | |
| | miRNA expression | 0.45 | 0.26–0.80 | 5.09E-03 | | | |
| CHOL | Age | 2.24 | 0.54–9.35 | 0.26 | | | |
| | Tumor grade | 0.99 | 0.21–4.65 | 0.99 | | | |
| | Tumor stage | 2.27 | 0.61–8.52 | 0.2 | | | |
| | miRNA expression | 0.09 | 0.01–0.79 | 7.59E-03 | | | |
| LUAD | Age | 0.98 | 0.56–1.69 | 0.93 | | | |
| | Tumor grade | | | | | | |
| | Tumor stage | 3.02 | 1.71–5.32 | 5.95E-05 | 2.74 | 1.55–4.87 | 5.56E-04 |
| | miRNA expression | 0.48 | 0.27–0.86 | 0.01 | 0.55 | 0.31–0.97 | 0.04 |
| BRCA | Age | 1.93 | 0.94–3.95 | 0.07 | | | |
| | Tumor stage | 4.04 | 1.99–8.19 | 2.77E-05 | | | |
| | miRNA expression | 0.51 | 0.25–1.06 | 0.06 | | | |

[a]HR hazard ratio.
[b]CI confidence interval.
[c]Log-rank test P.
[d]Wald test P.
[e]Histological grade.
[f]Early stages (I and II) vs. late stages (III and IV).
[g]Sample-wise average expression of miR-145-5p, miR-145-3p, miR-30a-5p, and miR-30a-3p. Samples were stratified into lowest quartile and highest quartile.

identified recurrent cancer-associated miRNA 5p/3p–pathway relationships. Determining recurrence of miRNA-pathway associations across cancer types itself is a powerful approach because it detects results that are reproducible in different cancers. The reproducible results that we observed in both analyses most likely indicate true biological events and minimize false-positives[46–48]. Taken together, these two massive-scale data analyses allowed us to pinpoint miRNA 5p/3p pairs whose dysregulation may consequently drive pathway/systems-level changes to favor cancer cells for survival, growth, and progression, referred to as phenotypic plasticity in cancer[49].

In both the analyses evaluating miRNA 5p/3p cooperativity, miR-145 and miR-30a appeared in the top prediction results. Their expression showed a negative impact on the viability/ growth of diverse cancer cell lines and they independently elevated cell cycle regulating pathway genes in 2/3rd of the cancer types in which they were downregulated. We determined that these miRNA-pathway gene associations were independent from the impact of DNA copy number and methylation changes and are consistent across the 14 cancer types using a robust rank product statistic[42,43]. With high statistical power and follow-up experiments in every step of the analysis, there is high confidence in our results.

Analysis of 531 heterogeneous normal tissues from 14 TCGA cancer types revealed that individual strands of miR-145 and miR-30a had 23-fold higher expression than the threshold to detect high-confidence miRNA[4,50]. In fact, miR-145-5p, miR-30a-5p, and miR-30a-3p are among the top 30 miRNA that constitute, on average, 90% of all miRNA expression across the normal tissues in TCGA[21]. Our study examined whether individual strands of high-confidence miR-145 and miR-30a can independently suppress the genes in crucial tumorigenic processes. From the miRNA perturbation profiles in LUAD (miR-145 and miR-30a) and ESCA cell lines (miR-145), we determined that elevated expression of both 5p and 3p strands independently suppressed an enriched number of cell cycle-associated pathway

genes. We observed consistent downregulation of miR-145 5p/3p pairs in TCGA, as well as in an independent LUAD patient cohort[34]. Previously, it was reported that decreased levels of miR-145-5p and miR-145-3p increase proliferation and movement of LUAD[37], LUSC[38], and BLCA[39] cells. In support of this, we showed that forced expression of both strands of miR-145 decreased lung adenocarcinoma cell growth and migration. Expression and functions of miR-30a 5p/3p pairs were unknown, but our data showed reduced levels in a panel of LUAD cell lines and in our own lung cancer patient samples. We also determined miR-30a-5p and 3p forced expression consequently reduced lung cancer cell proliferation and movement. Collectively, our data show that both strands of a miRNA can be abundantly expressed across tissue types, and they are capable of modulating tumorigenic processes independently and cooperatively.

Our comprehensive data analysis highlights that miRNA 5p/3p pairs, which do not have identical mRNA-targeting seed sequences, coordinately regulate the same pathways by targeting different genes in the pathway or the same genes at different sites. We determined from a pool of miR-145-5p/3p–pathway associations that only ~10% of the pathway genes were potentially co-regulated by both strands. It was slightly higher (28%) for miR-30a. Therefore, simultaneous downregulation of these miRNA might impact a larger spectrum of genes in specific biological processes compared to the single strand alone. These results also highlight a potential caveat of retroviral or transgene miRNA overexpression data, as both strands will be overexpressed, but typically only levels and targets of one strand are evaluated. Our data confirmed cooperative effects in the modulation of lung cancer cell growth and movement when both strands of miR-30a and miR-145 were perturbed simultaneously. Our results also expose cooperativity between all four strands of miR-30a and miR-145, since the four strands together had a larger impact on tumor cell proliferation and movement than both strands of either miR-30a or miR-145 alone. Furthermore, coordinated downregulation of miR-145 and miR-30a 5p/3p pairs

were an independent prognostic factor, resulting in patients with worse overall survival in multiple cancer types. Collectively, our multiple approaches with corresponding verification identified a new clinically relevant pan-cancer miRNA signature that had remained unidentified due to a lack of understanding of 5p and 3p strand-mediated cooperativity.

## Methods

**Analysis of TCGA transcriptomic data**. We extracted and processed miRNA-seq and mRNA-seq expression profiles of 14 TCGA cancer types. For each cancer type, we selected miRNA or mRNA for the downstream analysis if at least 50% of the samples had a normalized expression value ≥1. We used normalized expression profiles, for miRNA-seq-Reads Per Million Mapped Reads and for mRNA-seq-Reads Per Kilobase Million Mapped Reads, to measure miRNA-miRNA and miRNA–mRNA expression associations using Spearman's rank correlation method. We used raw read counts to measure differential expression of miRNA and mRNA. We employed the R/Bioconductor package edgeR for differential expression analysis[51]. In edgeR, the data were normalized based on negative binomial distribution. Differential expression of miRNA or mRNA between tumor and corresponding normal samples was assessed by estimating an exact test P-value, which is similar to the Fisher's exact test. The results were further adjusted using the Benjamini–Hochberg (BH) multiple testing correction method[23]. The mRNA were regarded as significantly differentially expressed in cancer compared with the normal tissue if they had at least 2-fold-change with BH adjusted $P < 0.05$. For miRNA, we used a cutoff of 1.5-fold-change with BH adjusted $P < 0.05$ since miRNA with 1.5-fold-change are reported to have significant impact on cellular processes[52].

**Prediction of cancer-specific dysregulated miRNA–mRNA pairs**. We extracted genome-wide predicted targets of miRNA from TargetScan (version 7.1)[53] and our previously developed miRNA-target prediction tool TargetMiner[54]. Additionally, we extracted high-confidence miRNA-targets from miRTarBase database that curated experimentally verified miRNA-gene interactions[55]. We took the union set of these three. For each cancer type, we predicted miRNA-mediated suppression of potential target genes by employing the following strategy: if the miRNA were significantly downregulated (at least 1.5-fold-change with BH adjusted $P < 0.05$), the predicted target genes should be significantly upregulated (at least 2-fold-change with BH adjusted $P < 0.05$) or vice-versa. Furthermore, the miRNA-gene pairs were required to be inversely correlated. This strategy identified cancer-specific miRNA-mediated potential suppression of the critical genes, which likely minimizes false-positives and captures the most important regulations in the context of cancer.

**Identification of cell survival/growth modulating miRNA**. We extracted genome-scale RNAi screening data on 712 cancer cell lines (Release- DEMETER2 Data v5) from Depmap database. From this database we also extracted genome-scale CRISPR screening data on 563 cancer cell lines (Release- Depmap Public 19Q2). Dependency scores for ~17,000 genes in a given cell line were determined by the algorithm DEMETER2[25] for RNAi and the algorithm CERES[26] for CRISPR. For the screening data, a lower score or rank means a given cell line is more likely to be dependent on that gene for its growth/survival. For individual cell lines, we ranked the ~17,000 genes based on their viability/growth scores determined from RNAi or CRISPR-mediated knockdown screens[24–26]. Cell lines in the RNAi or CRISPR screening data that originated from one of the 14 TCGA primary tumor sites we studied here were initially selected for this analysis. However, among the 14 cancer types, 5 were excluded from the analysis due to an insufficient number of cell lines (range 0–4). Finally, we evaluated 251 RNAi and 177 CRISPR screened cell lines, respectively (290 unique cell lines) that originated from 9 primary tumor sites. Our approach integrated (a) RNAi or CRISPR screening data of cell lines originating from a primary tumor site $i$, (b) predicted miRNA-target networks that were dysregulated and had significant (BH adjusted $P < 0.05$) inverse Spearman's rank correlations in $i$, and (c) gene set enrichment analysis (GSEA)[27] method, embedded into R package WebGestaltR[28] (Fig. 1d). Unlike traditional GSEA that uses curated pathways as gene sets, here we used targets of individual mature miRNA as the gene set. For a given miRNA and cell line pair, we implemented the GSEA approach on the ranked gene list and the miRNA-target gene set. The gene sets with significantly (FDR < 0.1) negative NES indicate they were overrepresented at the bottom of the ranked gene list.

**miRNA 5p/3p−pathway associations**. Differentially expressed genes that were potentially regulated by either 5p or 3p or both strands of a precursor miRNA in a specific cancer type were selected to conduct pathway enrichment analysis using the canonical KEGG pathway database[29], which is embedded into WebGestaltR, an R package of the web-server WebGestalt[28]. The pathways significantly enriched (BH adjusted hypergeometric test $P < 0.05$) with the predicted target genes were filtered as follows to obtain the high-confidence 5p/3p–pathway combinations. First, we examined whether the predicted miRNA-target genes had significantly stronger (BH adjusted $P < 10^{-3}$; Wilcoxon rank-sum test) inverse associations,

measured by the Spearman's rank correlations, compared to the 10,000 randomly selected miRNA-gene pairs. The genes in the randomly generated data set may or may not have predicted miRNA-binding sites of the corresponding miRNA. Second, we determined what proportion of the genes in a pathway were predicted to be targeted by the 5p or the 3p strand. Absolute difference between these two proportion values were taken where the lower the score indicates the greater the 5p/3p pair mediated coordinated regulation of the pathway. We selected those 5p/3p pair−pathway combinations that had the score <0.5. Third, we examined frequency of the 5p/3p pair−pathway association across the cancer types in which the miRNA 5p/3p pair was concordantly dysregulated. The higher the scores indicate the pathways are more recurrently dysregulated potentially due to concordant dysregulation of the regulating 5p/3p pairs.

**Multivariate regression analysis**. We collected processed DNA methylation and copy number profiles of 14 TCGA cancer types from the UCSC Xena database. To determine whether mRNA ($Y$) and its predicted regulating miRNA ($x_{miR_m}$) expression association in the $n$ tumor samples of a given cancer type was independent of DNA methylation ($x_{DM}$) and copy number ($x_{CNV}$) change events, we performed the following multivariate regression analysis:

$$Y_s = \beta_0 + \beta_{DM}x_{DM,s} + \beta_{CNV}x_{CNV,s} + \beta_{miR_m}x_{miR_m,s} \quad , s = 1, \dots, n$$

here $\beta_0$ is the intercept. $\beta_{DM}$, $\beta_{CNV}$, and $\beta_{miR_m}$ are the regression coefficients that indicate the association strength between RNA level expression change of the given gene with DNA methylation, copy number, and miRNA expression changes, respectively. For a specific cancer type, we used BH adjusted regression $P < 0.05$ as the cutoff to select significantly associated miRNA–mRNA pairs.

**Predicting recurrent miRNA–mRNA associations across cancers**. The miRNA–mRNA regression coefficients obtained from 14 cancer types were collated into a matrix format and input into the RankProducts function in the R package RankProd[42]. The coefficient values were ranked from most negative to most positive, and most positive to most negative. The rank product statistic, described by Breitling et al.[43], was then used to identify recurrence of miRNA–mRNA associations with statistical significance across the cancer types. The nominal P-values obtained from this analysis were further corrected using BH multiple test correction method. Consistent inverse associations across the 14 cancer types with BH adjusted $P < 0.05$ were selected as miRNA–mRNA association recurrence.

**Cell culture and transfections**. Lung cell lines were cultured in RPMI-1640 media supplemented with 10% FBS and penicillin/streptomycin (H460, H2087, H2110, H441, PC9, and H1299) and with additional L-glutamine (Beas2B, A549, H1993, and H1435). All cell lines were verified to be mycoplasma negative. Lipofectamine 2000 (Invitrogen) was used to transfect A549 and H1993 lung cancer cell lines with either 25, 50, or 100 nM miR-30a-5p, miR-30a-3p, miR-145-5p, or miR-145-3p miRNA mimic, negative control RNA mimic, or a combination (Dharmacon), according to the manufacturer's protocol using the 6-well plate format.

**RNA isolation and qRT-PCR**. RNA was isolated from cells and tissue using TRIzol (Ambion) according to the manufacturer's protocol with one modification. To enrich for miRNA, the isopropanol incubation step was performed overnight at −20 °C. To measure expression of miR-30a-5p and miR-30a-3p by qRT-PCR, TaqMan MicroRNA Assays (Applied Biosystems) were performed in triplicate. miRNA expression was normalized to the small RNA *RNU6B* as an endogenous control and is presented as $2^{-\Delta Ct}$.

**RNA-sequencing data analysis**. Triplicate samples were harvested 72 h after transfection of miR-30a-5p or miR-30a-3p miRNA mimic or control RNA into A549 cells. RNA-seq profiles, generated from Illumina Novaseq 6000 system, were obtained from Novogene (https://en.novogene.com/). Sequence mapping and quantification were performed using the methods STAR (V2.5)[56] and HTSeq (V0.6.1)[57], respectively. For sequence mapping, Human reference genome version hg19 was used. The read counts were imported into edgeR for differential expression analysis[51]. In edgeR, data were normalized based on negative binomial distribution. Differential expression of genes between miR-30a-5p or miR-30a-3p mimic and negative control RNA samples was assessed by estimating an exact test P-value. The results were further adjusted using the BH multiple testing correction method.

**Cell proliferation assay**. Lung adenocarcinoma cells (A549 and H1993) were transfected with individual miRNA mimics (25, 50, or 100 nM), a combination of miRNA mimics, and/or negative control RNA (equal amounts of RNA were transfected for each experiment). Cells were trypsinized, and 2500 cells/well were placed into 96-well plates, in quadruplicate. Once cells attached, the 0 h time was measured by MTT (Sigma) assay (approximately 24 h after transfection). MTT assays were performed every 24 h thereafter. Absorbance was measured at 562 nm.

**Colony formation assay**. Lung adenocarinoma cells (A549) were transfected with individual miRNA mimics (25 nM), a combination of miRNA mimics, and/or negative control RNA (equal amounts of RNA were transfected for each experiment). Twenty-four hours after transfection, cells were trypsinized, counted, and seeded at low density (300 cells/well) into 6-well plates, in triplicate. After 12 days in culture, colonies were stained using crystal violet and were counted (≥50 cells defined a colony) using an inverted microscope.

**Cell migration assay**. Lung adenocarcinoma cells (A549) were transfected with miRNA mimics (25 or 50 nM) separately or in combination and/or negative control RNA (equal amounts of RNA were transfected for each experiment). Twenty-four hours later, $4 \times 10^4$ cells were resuspended in serum-free RPMI-1640, placed into migration chambers (8 μm pores, BioCoat Insert), and transwell migration assays were performed, as we reported previously[47,58]. Specifically, inserts were stained 18 h later using Siemens Diff-Quick staining set and protocol and blinded. Migrated cells per field were counted (at least four independent fields per sample) using an Olympus CKX53 inverted microscope (10x objective).

**Patient sample acquisition**. De-identified frozen patient samples of non-small-cell lung adenocarcinoma (stages 3 and 4) and normal lung were obtained from the Vanderbilt University Medical Center Lung Biorepository. All patient samples were from surgical resections and were banked with patient consent. All samples were evaluated by a board-certified pathologist and were either determined to be >80% tumor (for tumor samples) or to lack any precancerous lesions (for normal samples) via H&E-stained sections.

**Statistics**. Unpaired, two-tailed $t$-tests were used when comparing two groups and $P$-values of <0.05 were considered statistically significant. Differential expression analysis of miRNA and mRNA between normal and cancer tissues from TCGA and the differential expression analysis of mRNA between miR-30a-5p and -3p mimics and negative control RNA in the A549 cell line were performed with edgeR[51]. Other statistical tests are indicated explicitly. For the biological assays (MTT, colony, migration, and qRT-PCR), data are presented as mean values ± SEM.

**Reporting summary**. Further information on research design is available in the Nature Research Reporting Summary linked to this article.

## Data availability

Data supporting the findings of this study are available within the paper and its Supplementary Information files. The RNA-sequencing data are available from Gene Expression Omnibus (GEO) and NCBI Sequence Archive (SRA) databases, GSE142695 and SRP238971. All other publicly available data referenced herein can be retrieved from GEO (https://www.ncbi.nlm.nih.gov/gds/), TCGA (https://portal.gdc.cancer.gov/), and the following websites- Depmap (https://depmap.org/portal), Xena (https://xena.ucsc.edu/), WebGestalt (http://www.webgestalt.org/). The Source Data for Figs. 3a, c–e, 4b, c, 5e, and 6a–f, and Supplementary Figs. 5a, b, 6 and 8a–c are provided as a Source Data file.

## Code availability

We make extensive use of previously published algorithms as described in the Methods and Results section. Additional code used in this study is available upon request.

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

## Acknowledgements
We thank Shanequia Jackson for technical assistance and members of the Eischen lab for helpful discussions. Support for this study was provided by NCI R01CA177786 (C.M.E.), the Pellini Foundation (C.M.E.), the Herbert A. Rosenthal, MD endowed chair (C.M.E.), NCI Cancer Center core grant P30CA056036 that supports the MetaOmics core facility and the Sidney Kimmel Cancer Center.

## Author contributions
R.M. and C.M.E. designed the study; R.M. performed all the bioinformatics analyses; W.J. performed the initial TCGA RNA-seq data analyses; C.M.A. and E.G. performed the biological experiments and evaluated these data; R.M. and C.M.E. evaluated all the data and wrote the manuscript; and all authors read and approved the manuscript.

## Competing interests
The authors declare no competing interests.
