## [Peer Review File · Nature Communications]

Reviewers' comments:

Reviewer #1 (Remarks to the Author):

This manuscript by Mitra et al. mostly describes an analysis of publically available databases to identify pairs of microRNAs transcribed from the same precursor and showing the same function. The authors then chose miR-30a and miR-145 and demonstrated the tumor suppressor function of these microRNAs.

Critique.

1. There are multiple publications about tumor suppressor function of all 4 microRNAs: miR-30a-3p, miR-30a-5p, miR-145-3p and miR-145-5p. That also includes publications showing that both isoforms of miR-30a and miR-145 are tumor suppressors. So this manuscript has a very limited novelty. Seems like the authors did that extensive analysis just to find what was mostly already known. Did they find any other similar cases where such function was previously unknown? Can they functionally study those cases?

2. The authors carried out an extensive analysis of publically available databases. Did they experimentally confirm any of these data (except for miR-30a)? In my opinion this manuscript is more suitable for a journal that focuses on bioinformatics, as most of other experimental data is only a conformation of previously published materials.

Reviewer #2 (Remarks to the Author):

Mitra et al. presented a pan-cancer analysis to show that different strands of the same miRNA work coordinately modulate oncogenic pathways and regulate cell survival across cancer types. They further focused on the 5p/3p strands of miR-145 and miR-30a and provided independent experimental validation about their downregulation, their effects on cell cycle and the cooperative effects. Finally they showed the expression signals of both strands of miR-145 and miR-30a represent a robust prognostic signature across cancer types. Overall, I think this is a nice study. The author integrated both patient cohorts and cell-line perturbation data to identify key miRNAs and the observed patterns are robust across multiple datasets and multiple cancer types. Although the role of miRNA regulation in cancer has been recognized for a long time, the cooperative effects of miR-5p/3p are less appreciated, which is the major/ novel contribution of this study.

1. For the patient survival analysis, the author averaged the expression of miR-145-3p/5p and miR-30a-3/5p. What are the results if only using miR-145-3p/5p or miR-30a-3p/5p or only a single miRNA separately? A side-by-side comparison will help assess how much information is gained by considering both 3p/5p?

2. P9, for the other 19 miRNA 5p and 3p pairs who coordinately target the same pathways, do their

expression show similar prognostic power as miR-145/30a demonstrated?

3. Abstract is not that informative. At least the names of the two key miRNAs miR-145/30a should be included.

4. When multiple miRNAs were mentioned, it should be "miRNAs"

5. Fig. 5, B and D, "10 \times -09" should be "10 \times -9"

6. Fig6, since multiple cancer types were assessed, FDR should be reported

Response to Reviewers

We thank the Reviewers for their time and comments with regards to our manuscript. We have addressed each Reviewer comment point-by-point below and have altered the text accordingly (edits to the text are in red). We have also added additional data (see Figures 6D-F, Supplementary Figures 8B and 8C, and Supplementary Table 9) to our manuscript in response to Reviewer comments. These data further support our conclusions and add additional depth and novelty to our manuscript.

Reviewer 1:

1. There are multiple publications about tumor suppressor function of all 4 microRNAs: miR-30a-3p, miR-30a-5p, miR-145-3p and miR-145-5p. That also includes publications showing that both isoforms of miR-30a and miR-145 are tumor suppressors. So this manuscript has a very limited novelty. Seems like the authors did that extensive analysis just to find what was mostly already known. Did they find any other similar cases where such function was previously unknown? Can they functionally study those cases?

We apologize the many novel aspects of our manuscript were not clear to this Reviewer. From one of the largest comprehensive high-throughput cancer data analyses to-date, we show for the first time recurrent pan-cancer cooperativity of specific miRNA 5p and 3p strands, previously unknown associations between miRNA 5p/3p pairs and oncogenic pathways, a novel miRNA signature significantly associated with patient survival across multiple cancers, a novel approach to determine miRNA and cancer cell survival association, and a novel integrative bioinformatics approach that can now be used by other scientists around the world to determine cooperativity of other genes in cancer and other diseases. We also validated these data with multiple approaches, including biological functional studies.

Also, we carefully reviewed the literature before choosing miR-30a and miR-145 to further investigate. This is, in part, because miRNA are reported to have different functions, including the opposite functions, in different cell and cancer types, and miR-30a and miR-145 are no exception to this. For example, miR-30a is reported to be oncogenic in some cancers and tumor suppressive in other malignancies, although only one strand was usually evaluated to make this conclusion and some studies lacked sufficient scientific rigor (reviewed in Yang et al *Cellular Physiology and Biochem.* 2017). In addition, it is reported that the 5p miRNA strand can have a different function or the opposite function from the 3p strand or that only one strand is functional (e.g., Ren et al. *Cell Death & Dis.* 2018, Chou et al. *Anticancer Res* 2018, Almeida et al. *Gastroenterology* 2012, Zhang et al. *Nat. Commun.* 2019, Gregory et al *Cell* 2005). Therefore, it was not clear prior to our analysis that there would be any miRNA 5p/3p pairs that would cooperatively regulate the same pathway across multiple cancer types and if any were identified, whether they would impact patient survival in multiple different cancers. However, our novel computational biology approaches identified 21 miRNA whose 5p/3p strands cooperatively regulated the same pathways. Each of these 21 miRNA had been previously investigated at some level in at least one cancer type, but none had been investigated for pan-cancer cooperativity. Most notably, our pan-cancer analyses identified core critical miRNA (miR-30a-3p, miR-30a-5p, miR-145-3p and miR-145-5p) whose both strands impact multiple cancers through cooperatively targeting the same pathways and make up a miRNA signature that has patient prognostic power in many cancers. Thus, our novel approaches with corresponding verification identified a clinically relevant pan-cancer signature miRNA that had remained unidentified due to a lack of understanding of 5p and 3p strand-mediated cooperativity. Additionally, our manuscript provides a new computational biology

approach for determining cooperativity of critical genes in cancer and other diseases that other scientists can now use in their studies. Together, our novel computational biology approaches and validation significantly expand understanding of miRNA in cancer by revealing that strands of specific miRNA work together to regulate core pathways in multiple cancers and that these impact cancer cell survival and patient survival. We have also performed additional experiments and analyses in response to both Reviewers to add even more novelty to our revised manuscript (see descriptions below).

- 1) In response to Reviewer 1, we have performed additional biological experiments combining all four miRNA strands (miR-30a-5p, miR-30a-3p, miR-145-5p, and miR-145-3p) to evaluate the combined effects of these four miRNA in lung adenocarcinoma. Our new data show that simultaneous changes in the abundance of all four members of our signature miRNA exert a significantly stronger effect on the growth, colony formation, and movement of lung cancer cells as compared to individual miR-30a or miR-145 5p/3p pairs. With this result, for the first time, we report functional cooperativity of four strands of 2 miRNA, which formed a novel signature miRNA, on lung cancer cell growth, colony formation, and motility. We added these new data to the revised manuscript as Figures 6D-F and Supplementary Figures 8B and 8C and text describing them to the results on page 16.
- 2) In response to Reviewer 2, we have added data showing a significantly ($P=2.51 \times 10^{-3}$; Wilcoxon rank-sum test) stronger association between poor overall survival and down-regulation of all four strands of the signature miRNA compared with the association between reduced overall survival and down-regulation of single strands or both strands of miR-145 or miR-30a, which formed the miRNA signature to page 17. These data are now in Supplementary Table 9. These results support the clinical relevance of concurrent down-regulation of the identified miRNA signature in many cancers.
- 3) In response to Reviewer 2, we have assessed whether other miRNA we identified, which cooperatively regulate the same critical pathways (Supplementary Table 6), are associated with significantly reduced patient survival. We determined that none had the equivalent or superior survival association as that of the miR-145/30a 5p/3p combination, indicating this signature miRNA is unique and has pan-cancer clinical relevance. We have added text describing these results to page 17.

2. The authors carried out an extensive analysis of publically available databases. Did they experimentally confirm any of these data (except for miR-30a)? In my opinion this manuscript is more suitable for a journal that focuses on bioinformatics, as most of other experimental data is only a conformation of previously published materials.

We have verified that *Nature Communications* publishes purely computational biology papers that report novel computational biology approaches and previously unexplored high-confidence prediction results. Our manuscript reports a novel approach to determine miRNA and cancer cell survival association and a novel integrative bioinformatics approach that can now be used by computational biologists around the world to determine cooperativity of other genes in cancer and other diseases. However, what elevates our manuscript above the many purely computational biology papers is that we also biologically validated results from our computational biology approaches. Our data showing the cooperative impact of both strands of miR-30a compared to individual strands in lung adenocarcinoma was unknown. However, there was a paper on miR-30a-5p and 3p showing each impacting Wnt signaling in esophageal squamous cell carcinoma, but

they only evaluated each strand separately (Qi et al. *World Journal of Gastroenterology* 2017). Our experiments with a different cancer (lung adenocarcinoma), testing individual 5p and 3p strands and both strands together in cooperativity tests in biological assays and through integrative systems biology approaches that utilized large-scale CRISPR and RNAi screens, and pathway level regulations, showed that miR-30a-5p and 3p work cooperatively to inhibit cancer cell proliferation and migration. Also, to provide additional novelty to the biological verification aspect of our manuscript since the first submission, we have performed additional experiments to test the cooperation of both strands of miR-30a and both strands of miR-145 together (see Figure 6D-F and Supplementary Figures 8B and 8C), which has not been previously reported. The results show that combining miR-30a-5p, miR-30a-3p, miR-145-5p, and miR-145-3p together significantly inhibited lung adenocarcinoma proliferation, colony formation, and migration compared to miR-30a 5p/3p or miR-145 5p/3p pairs alone. Moreover, effects from the four miRNA strands together were evident at half the concentration than that of only one 5p/3p pair, providing additional evidence of functional cooperativity among these four miRNA strands. These data provide novel results and further highlight the importance of the miRNA signature (concurrent down-regulation of miR-30a-5p, miR-30a-3p, miR-145-5p, and miR-145-3p) associated with pan-cancer patient survival. Together, our miRNA cooperativity results are highly significant, novel, and clinically meaningful.

Reviewer #2:

Overall, I think this is a nice study. The author integrated both patient cohorts and cell-line perturbation data to identify key miRNAs and the observed patterns are robust across multiple datasets and multiple cancer types. Although the role of miRNA regulation in cancer has been recognized for a long time, the cooperative effects of miR-5p/3p are less appreciated, which is the major/ novel contribution of this study.

We thank the reviewer for recognizing this particularly novel aspect of our study.

1. For the patient survival analysis, the author averaged the expression of miR-145-3p/5p and miR-30a-3/5p. What are the results if only using miR-145-3p/5p or miR-30a-3p/5p or only a single miRNA separately? A side-by-side comparison will help assess how much information is gained by considering both 3p/5p?

As recommended by the Reviewer, we evaluated patient overall survival association with the expression of an individual miRNA strand or the average expression of both the strands of miR-145 or miR-30a. We observe a significantly ($P=2.51 \times 10^{-3}$; Wilcoxon rank-sum test) stronger association between poor overall survival and down-regulation of all four strands of the signature miRNA compared with the association between reduced overall survival and down-regulation of single strands or both strands of miR-145 or miR-30a. We thank the reviewer for recommending this analysis, which provides additional clinical relevance of concurrent down-regulation of the identified signature miRNA during tumorigenesis in multiple cancer types. A side-by-side comparison is now shown in Supplementary Table 9.

2. P9, for the other 19 miRNA 5p and 3p pairs who coordinately target the same pathways, do their expression show similar prognostic power as miR-145/30a demonstrated?

We have assessed whether other miRNA we identified, which cooperatively regulate the same

pathways (Supplementary Table 6), are associated with significantly reduced patient survival. We determined that none had the equivalent or superior survival association as that of the miR-145/30a 5p/3p combination, indicating this signature miRNA is unique and has pan-cancer clinical relevance. We have added text describing this to page 17.

3. Abstract is not that informative. At least the names of the two key miRNAs miR-145/30a should be included.

We revised the abstract adding in miR-145 and miR-30a and rewording several sentences while keeping within the 150 word limit.

4. When multiple miRNAs were mentioned, it should be “miRNAs”

In our experience, copy editors at different journals have different requirements in regards to plural miRNA. In our manuscript, we used miRNA as both singular and plural, since DNA and RNA are both singular and plural and fit the grammatical requirements of most journals.

5. Fig. 5, B and D, “10 \times -09” should be “10 \times -9”

We removed the zero before the number in Fig 5B and 5D, as recommended.

6. Fig 6, since multiple cancer types were assessed, FDR should be reported

We have modified Figure 7 (previously Figure 6) and now report both nominal p-values and FDR.

We again thank the Reviewers for their comments. With the added analyses/experiments and edits, our manuscript is significantly improved from the first submission. Our large scale bioinformatic results and experimental validation continue to show specific miRNA 5p/3p strands mediate cooperative regulation of tumorigenic processes and pathways. Furthermore, we used a novel integrative bioinformatics approach that can now be used by other scientists to determine cooperativity of genes and we identified a novel miRNA signature that has significant clinical relevance across many cancer types. We believe that due to the scale of our study and the significance of the results, our manuscript is appropriate for *Nature Communications*.

REVIEWERS' COMMENTS:

Reviewer #1 (Remarks to the Author):

The authors provided some more information, but still the main conclusions of this manuscript are about tumor suppressor function of these four microRNAs. And this was already described in multiple publications.

Reviewer #2 (Remarks to the Author):

The authors have addressed my concerns.

Response to Reviewers*REVIEWERS' COMMENTS:*Reviewer #1:

The authors provided some more information, but still the main conclusions of this manuscript are about tumor suppressor function of these four microRNAs. And this was already described in multiple publications.

We appreciate this reviewer for reviewing our manuscript. However, there is a misunderstanding of the main conclusions of our manuscript. The main conclusions of our manuscript are that we identified specific miRNA whose 5p and 3p strands cooperate to regulate core cellular pathways across multiple cancer types even though they have different targeting sequences and they are a novel miRNA signature significantly associated with patient survival in multiple cancer types. Secondly, our manuscript is also the first to report cooperativity of both strands of two miRNA in regulating cancer cell growth and movement. Thirdly, our manuscript also reports a novel integrative bioinformatics framework that scientists around the world may now implement to predict cooperativity of gene regulators in different contexts and diseases.

Reviewer #2:

The authors have addressed my concerns.

Thank you for your response.

We appreciate both the reviewers for their time in reviewing our manuscript. Their valuable suggestions enhanced the overall quality of our manuscript.